# Unconventional Applications of Superconducting Nanowire Single Photon Detectors

**DOI:** 10.3390/nano10061198

**Published:** 2020-06-19

**Authors:** Tomas Polakovic, Whitney Armstrong, Goran Karapetrov, Zein-Eddine Meziani, Valentine Novosad

**Affiliations:** 1Physics Division, Argonne National Laboratory, Argonne, IL 60439, USA; tpolakovic@anl.gov (T.P.); warmstrong@anl.gov (W.A.); zmeziani@anl.gov (Z.-E.M.); 2Department of Physics, Drexel University, Philadelphia, PA 19104, USA; goran@drexel.edu; 3Department of Materials Science and Engineering, Drexel University, Philadelphia, PA 19104, USA; 4Materials Science Division, Argonne National Laboratory, Argonne, IL 60439, USA

**Keywords:** superconductivity, nanowires, particle detectors, photon detectors, quantum detectors

## Abstract

Superconducting nanowire single photon detectors are becoming a dominant technology in quantum optics and quantum communication, primarily because of their low timing jitter and capability to detect individual low-energy photons with high quantum efficiencies. However, other desirable characteristics, such as high detection rates, operation in cryogenic and high magnetic field environments, or high-efficiency detection of charged particles, are underrepresented in literature, potentially leading to a lack of interest in other fields that might benefit from this technology. We review the progress in use of superconducting nanowire technology in photon and particle detection outside of the usual areas of physics, with emphasis on the potential use in ongoing and future experiments in nuclear and high energy physics.

## 1. Introduction

Superconducting nanowire single photon detectors (SNSPD) have, since their initial discovery [1], found many applications in fields of nanophotonics and quantum communication. Metrics like sub-20 ps timing jitter [2], nearly 100% quantum efficiency up to IR wavelength [3,4] and count rates as high as $10^{9}$ counts/s with effectively zero dark counts [5] make them the go-to choice in many applications, including LIDAR systems [6], quantum teleportation [7], quantum key distribution [8], optical quantum computing [9], and many others [10]. Many, if not all, of these applications leverage the unique capability of detecting individual photons with unprecedented timing resolution and low noise, but that doesn’t mean that this is the only area where SNSPDs can be used in.

One of the interesting potential applications of SNSPDs is in nuclear physics (NP) and high energy physics (HEP), where there is demand for new detector technologies for particle identification, calorimetry, and particle trajectory reconstruction. Currently, this area is dominated by semiconductor-based detectors but there is demand for new technologies at the intensity frontier [11], where the detectors need to operate at high rates and in harsh environments [12,13,14,15,16,17,18]. Superconducting nanowire detectors have already demonstrated superior timing characteristics and detection rates as photodetectors, they naturally operate in cryogenic conditions and there has been rapid progress in their capability to operate in strong magnetic fields [19], and they are capable of detecting more than just individual photon signals.

In this work, we review the additional capabilities of these devices in attempt to show that they can be a very attractive choice as a complementary or replacement technology for many of experiments outside their usual niche. In Section 2, a brief introduction to the detection principles in the conventional schemes is presented. Detection of multi-photon signals and photon counting in strong magnetic fields is discussed in Section 3. Section 4 will then focus on detection of signals from massive particles, beginning with keV-range energy ions and electrons, then MeV charged particles and will conclude with proposals on detection of neutrons.

## 2. SNSPD Concept Origins, Operation and Metrics

The basic principle behind the photo-detection in SNSPDs is excitation of quasi-electrons out of the superconducting ground state, an effect that has been first observed almost five decades ago in thin films of lead [20]. As was shown it the lead film experiments, the formation of the normal resistive state in the material cannot be explained purely by thermal effects. To fully account for the experimental results, one has to consider the non-equilibrium dynamics of the superconducting excited state, with hot quasi-particles at higher temperature than the Cooper pairs of the superconducting ground state. A more thorough analysis of physics of these processes not limited to applications of detectors can be found in review of Shklovskij [21], while in this work, we restrict ourselves just to a quick review of the body of work related to detectors.

One of the first attempts at quantitative description of this phenomenon was using a two-temperature model [22], which describes the photoresponse of the superconductor hit by a (not necessarily a single photon) pulse of light. This model assumes that the photoresponse, which is macroscopically equivalent to change of the kinetic inductance of the film (inversely proportional to Cooper pair density), has time constant that is dominated by that of the quasi-particle relaxation time [23,24,25]. The electron temperature $T_{e}$ and phonon temperature $T_{p}$ is then described by a pair of linearized coupled heat-balance equations [26,27]:(1)$$\begin{array}{cc} c_{e}\frac{dT_{e}}{dt} & =-\frac{c_{e}}{\tau_{e-p}}\left(T_{e}-T_{p}\right)+P\left(t\right),\\ c_{p}\frac{dT_{p}}{dt} & =\frac{c_{e}}{\tau_{e-p}}\left(T_{e}-T_{p}\right)-\frac{c_{p}}{\tau_{\mathrm{es}}}\left(T_{p}-T_{0}\right), \end{array}$$
where $c_{e}$ and $c_{p}$ are electron and phonon specific heat, $T_{0}$ the heatsink (substrate) temperature, $\tau_{e-p}$ the average electron-phonon interaction time, P(t) the absorbed radiation power and $\tau_{\mathrm{es}}$ the time of phonon escape from superconductor to substrate. The solutions to these equations can be further related to the film resistance as [22]:(2)$$R\left(T_{e}\right)=R_{n}{\left(1+\exp\left(-4\frac{T_{e}-T_{c}}{\Delta T_{c}}\right)\right)}^{-1},$$
where $R_{n}$ is the normal state resistance and $\Delta T_{c}$ the transition width at a given biasing current. This model predicts characteristic time constants on the order of 20–30 picoseconds, which is agreement with experiments carried out on, e.g., NbN microbridges [28,29].

The spatial dependence of this process was originally studied in context of non-ideal quasi-one-dimensional superconductors by Skocpol, et al. [30], where the concept of a hot-spot, was first introduced into the topic. Hot-spot is a region of suppressed order parameter in the superconductor that can form and persist at temperatures lower than the superconducting critical temperature and was studied in context of cryotrons (bi-stable superconducting switching devices) [31,32,33,34]. This concept was further elaborated by Kadin, et al. [35] who approximated the Equation (Equation 1) by a diffusion equation:(3)$$Cd\frac{\partial T}{\partial t}=\kappa d\nabla^{2}T+\alpha \left(T_{0}-T\right),$$
where $C=c_{e}+c_{p}$ is the superconductor specific heat, κ total thermal conductivity of the quasi-particles, α the superconductor-substrate boundary thermal conductance and $T_{e}=T_{p}=T$ is the temperature of the superconductor. Solution to this equation predict single photon-induced hotspots with sizes on the order of 10–100 nm, which lead to one of the first proposals of the SNSPD devices [36].

These models are enough to qualitatively describe the detection process, which can be divided into multiple stages, that are roughly sketched in Figure 1: A very thin and narrow nanowire is maintained well below the superconducting critical temperature $T_{C}$ and is constant-current biased at current values close to critical currents (Figure 1a). Absorption of a single photon with energy much higher than the superconducting energy gap ℏω≫2Δ≈2meV will lead to excitation of two quasi-electrons with high kinetic energies, and those will begin to inelastically scatter with other quasi-particles in the system. This forms the initial hot-spot [36,37,38,39,40] (Figure 1b). The approximate time-scales associated with this process are on the order of 10 ps [22] (dictated by the electron-phonon scattering times). The next stage of the detection process is the expansion of the hot-spot (Figure 1c). First, the quasi-particles further multiply due to inelastic scattering and move diffusively outwards [41], towards the edges of the nanowire. The depletion of Cooper pairs leads to local reduction of the order parameter and redistribution of the current in the region [36]. If the nanowire is biased close to critical currents, the current density in the hotspot region rises above the critical values and the superconducting state vanishes completely creating a normal state region that bisects the superconducting wire (Figure 1d).

More recent experiments [42] and calculations [43,44,45,46] show that this process is also aided by Abrikosov vortices—While the potential barrier for a vortex entry might be too high in the equilibrium state, reduction of the order parameter also reduces the nucleation energy and individual vortices might enter from the wire edges, or vortex-anti-vortex pairs can form and separate. These vortices move across the wire width due to Lorentz force caused by the flowing current and cause voltage transients, further reduce the order parameter or cause phase slips [47,48,49,50,51] that can contribute to the device dark counts. As the section of the wire turns normal, a voltage drop is generated across the wire and this voltage spike is registered as a resistance spike which corresponds to a single photon detection event. In the last stage, the current bias is reduced, either passively through a parallel resistive shunt, or actively using electronically controlled current source and the hot spot shrinks and vanishes, preparing the nanowire for the next detection event (Figure 1e). The time constant associated with this last step could, in principle, be determined by how fast the electronic system can cool down to substrate temperature through interaction with the phonon subsystem—this would be then on the order of the phonon escape time $\tau_{es}$, which can have values from roughly $\tau_{e-p}$ to a few nanoseconds [22]. However, in reality this is primarily determined by the inductance of the device. Many superconductors used in SNSPD devices have sheet kinetic inductances as high as ∼50pH/□ [52], so a typical ∼$100\;\mu m^{2}$ area device can have kinetic inductance around 0.5 μH that leads to the observed time scales of the reset time on the order of 10 ns.

While the time from photon absorption to registration of a voltage pulse is approximately 10 ps, the important metric is the timing *jitter*, which is the statistical variation of this time delay (typically reported as FWHM of the distribution). The variation has two contributions: an extrinsic variation due to geometry of the wire, where there’s a different time delay depending on where along the length of the wire does the photon get absorbed (in a typical meander geometry, the wire length can reach a few mm) and jitter associated with the timing of the readout electronics [2]. Then there is the intrinsic variation that’s due to the probabilistic nature of the processes that lead to the detection event. Typical values of timing jitter of SNSPD detectors are around 15 ps and are dominated by the noise jitter of the readout [53], which can reduced down to 7 ps, a value due to geometric jitter [54]. On a short nano-bridge geometry, the intrinsic timing jitter of 2.7 ps [55] is reported, which is thought as the current fundamental limit to timing of the SNSPD devices.

As many of the applications proposed in this work will demand larger pixel sizes than ∼100 μm^2^, it might be worth elaborating on the current trends on this front: While not as common, detectors with pixel sizes of the order of 10,000 μm^2^ are being fabricated [56,57]. Alternatively, one can connect multiple small SNSPD pixel in parallel to create a large-area “superpixel” [58,59,60,61,62]. An additional benefit of constructing a detector pixel in this configuration is a better detection rate, because the total inductance of the superpixel is the harmonic mean of inductances of individual detectors. The tradeoff is in reduction of SNR of the detector (the voltage drop in inversely proportional to the number of pixels connected in parallel), but typical signals are on the order of a few mV, so this is not a fundamental problem unless the number of parallel pixels becomes too large. Lastly, the engineering problems associated with many-pixel readout are being focused on by many groups working on superconducting electronics and working arrays of as many as 64 pixels have existed for some time [63,64,65,66] and, coupled with various switching readout and multiplexing techniques [65,67,68,69,70,71,72], a kilopixel SNSPD array has been recently developed [73]. Specifically in the context of particle detection, the covered area can be further increased by increasing the dimensions wire itself—while this is generally not possible with photons, charged particles can create hotspots an order of magnitude larger (as will be discussed in Section 4). This would put nanowire-based particle detector (super)pixels comfortably in the mm-scale and make them competitive with conventional technologies also in this aspect.

While most of the previous exposition was not specific to any particular superconducting material, it might be worth closing the introductory section with a short discussion about materials commonly used in fabrication of the detectors. The first reported SNSPD was fabricated out of NbN [1], which is still a popular material to use because of ease of fabrication and comparatively high $T_{C}$ and robustness against chemical damage during fabrication or operation. Another commonly used materials from the family of nitrides are TaN [74,75] and NbTiN [52,76,77]. All three of these materials have similar properties in context of photon detection and decision is usually based on practical factors: Materials with smaller superconducting gap will have higher relative sensitivity to long wavelength photons. The tradeoff for better sensitivity in infra-red is lower critical current density and $T_{C}$, which translates into worse signal-to-noise ratio and need for better cryogenics. The main drawbacks of this family of materials is their sensitivity to chemical and physical conditions during thin film deposition, where substantial efforts have been made to achieve better consistency and process compatibility [78,79,80], and high reflectivity [81,82,83] that makes anti-reflective coatings or integration into photonic cavities necessary to achieve high quantum efficiency [4,84]. The second common class of SNSPD materials is that of amorphous superconductors, mostly popular for long wavelength photon detection. The amorphous nature of the materials allows for fabrication of extremely narrow and thin nanowires and they generally have smaller superconducting gaps than nitrides, which greatly enhances their IR sensitivity, as demonstrated with WSi [3], MoSi [85] or MoGe [86], with MoSi also having demonstrated good performance in UV [87]. The decision between the various amorphous materials is based on the balance of sensitivity, noise, temperature and jitter, as with the crystalline materials.

An interesting alternative to conventionally used materials are the high-$T_{C}$ superconductors (specifically cuprates), where alternative detection models are also proposed [88]. The obvious benefits of using this material the high critical temperature and upper critical magnetic field. The tradeoff in this case comes mostly from the side of technology, where growth of high-quality superconducting oxides is a complicated process. YBCO films, which are commonly proposed for high-$T_{C}$ SNSPDs, require highly specific substrates and growth at excessive temperatures [89,90,91,92] and have to be capped by a passivation layer to prevent degradation [89]. This limits application in detectors that need on-chip integration with other devices, but one should not discredit their use in detector concepts that do not require such degree of integration. The highest $T_{C}$ material has been demonstrated in use for SNSPDs is MgB_2_ [93,94], however, the material suffers from high sensitivity to oxygen and epitaxy [95], and single-photon detection has been observed only at temperatures lower than 20 K [96].

When deciding between the whole range of materials, it is important to consider the demands of the experiment. Nitrides have higher critical temperatures, which allows them operate even at liquid helium temperatures and they have smaller timing jitter [54,55] but they have lower sensitivity to long wavelength photons and are limited to smaller number of possible substrates. In context of NP and HEP experiments, there is also the additional consideration of effects of the environment. High magnetic fields will favor the polycrystalline materials with high vortex pinning, and devices made out of these materials currently perform better in these conditions [19,97]. Another option for magnetic field-tolerant devices would also be to fabricate them out of high-$T_{C}$ materials. Experiments expecting high radiation fields or high-energy particle bombardment should also tend towards use of the crystalline materials, specifically NbN, because Nb has comparatively low neutron capture [98] and scattering [98,99] cross-sections and the short screening length [100,101] makes it more robust against lattice defects.

## 3. Non-Standard Photo-Detection Techniques

### 3.1. Multi-Photon Detection and Multi-Layer Broadband SNSPDs

Most of the applications of SNSPDs are in single photon counting of faint signals in narrow spectral range and are discussed thoroughly in literature [10,102]. In this review, we want to highlight techniques that use these devices in different scenarios or configurations, and we start with the less known applications that still focus on detection of light, which is still of importance to NP and HEP experiments (e.g., detection of broadband light from Cherenkov radiation, scintillation or ionization).

First such application is number-resolved detection of multi-photon signals, which is commonly facilitated by visible light photon counters [103,104], transition edge sensors [105], or even Si avalanche photodiodes [106]. The benefit of using SNSPDs over the other technologies is the superior timing jitter and rate characteristics and simple readout scheme, while not suffering from the drawbacks of being confined to narrow temperature regions around room temperature or in the mK range. Typically, multi-photon resolution is achieved by having arrays of multiple SNSPDs connected in parallel and use a readout that sums inputs from this array, as demonstrated, e.g., in work of Mattioli, et al. [107]. However, one doesn’t need to limit themselves to SNSPD arrays to achieve multi-photon resolution, as individual SNSPD devices also have this capability in certain circumstances. These detection capabilities have been demonstrated by Cahal, et al. [108] in 2017 on standard meander devices made out of amorphous superconductors. Phenomenological models describing the electro-thermal feedback during the hot-spot formation [109,110] show that the voltage pulse signal due to multiple nearly simultaneous detection events (formation of multiple hot-spots within the time-span of nucleation and expansion stages of the detection process) has a rise time that is inversely proportional to the square root of total number of hot-spots. Using a simple inductor-resistor differentiating circuit, Cahal, et al. were able to demonstrate photon number resolution at 1550 nm wavelength using approximately 80 ps attenuated laser pulses, as shown in Figure 2.

Another technique, which is not yet standard in SNSPD applications, and could find another uses in the future is using multi-layer devices, with patterned superconducting layers separated by dielectric spacing layers [111,112,113,114] as shown in Figure 3. As the most popular superconductors used for SNSPD fabrication have reflectivity of more than 0.5 in most of the interesting wavelength range [115], one requires to integrate them into photonic cavities or integrate anti-reflective coatings, which turn them into inherently narrowband detectors. This can be alleviated by tuning the thickness of the dielectric spacing layers to increase absorption for different wavelengths in different superconducting layers and effectively making a broadband detector device, as was successfully demonstrated in NbN-based detectors [111,113]. Similar attempts were made using amorphous WSi as the superconducting material [112], but they didn’t achieve increase of detection efficiency as with the crystalline materials—something that is argued in the work to be due to fabrication defects and insufficiently low reduced working temperature $T/T_{C}$.

### 3.2. Detection in Strong Magnetic Fields

Before we move on to detection of particles, we should discuss the detection capabilities of SNSPDs in strong magnetic fields. There exist experiments which require high-rate and high-speed detection of photons that cannot be covered by conventional technologies.

One example from nuclear physics is the active polarized proton target [116] where an experiment measure the proton spin-polarizabilities through Compton scattering off a polarized proton target. The proton spin polarization is achieved via the dynamic nuclear polarization technique of a solid target held at cryogenic temperatures and in fields as high as 6 T. Within the solid target material the presence of nuclei, other than the polarized protons, causes a significant background, which can be eliminated by detecting the recoil proton within the target material. At the required cryogenic temperatures, charge carrier freeze-out depletes Si-based photomultipliers and causes serious performance degradation [117], often requiring a complicated optical photon collection scheme and scintillator doping of the target material [118].

Also, the presence of strong magnetic fields also leads to decrease of detection efficiency, especially when using photomultiplier tubes. The inefficiency is mitigated by placing the photodetectors outside of the cryogenic and high-field environment, either by increasing the physical distance [119], by including magnetic shielding [120], and by guiding the light towards the detector with optical waveguides or systems of mirrors. Doing this puts a strain on the material budget, induces insertion losses or attenuation in the medium, and increases timing uncertainty due to dispersion. Considerable R&D efforts are spent to achieve photon detection performance in at least 1 T [121].

Having detectors in close proximity of the active volume of the experiment, in many cases, would alleviate these problems. As SNSPDs are superconducting, operation in cryogenic environments doesn’t cause any degradation in their performance, the only open question was their performance in magnetic fields. Low-dimensional superconductors (such as superconducting nanowires) can persist in a Meissner state at fields higher than those for bulk superconductors [122,123], but large transport currents and the reduction of order parameter during the detection events will lead to reduction of the Abrikosov vortex nucleation energy barrier [124,125]. These vortices, whether they entered from the sample surface or formed by unbinding of vortex/anti-vortex pairs (the basic topological excitation in the 2D superconductors) will move across the sample under action of the Lorentz force from the transport current and dissipate heat on the order of ∼100meV per vortex crossing [125,126]. This can lead to excess dark counts in magnetic fields but also enhance detection of photons that would be too low-energy to trigger a detection event in a vortex-free superconductor [46,127]. Because of this direct impact of magnetic fields on the detection mechanism of SNSPDs, considerable work has been done on studying the effects of small fields, usually not more than 200 mT [128,129,130], which is not enough to meet the demands of many potential experiments. However, efforts in geometry optimization of SNSPD pixels [97,131] and progress in superconductor film deposition technology [78] allows SNSPDs to operate in fields as high as 0.5 T when applied perpendicular to pixel plane and 5 T for fields parallel to the pixels [19]. If needed, this values could be potentially increased, if geometry and material optimization were to be combined [131].

## 4. Particle Detection

Detection of energetic charged particles can be facilitated by two mechanisms, both of them sharing the underlying physics of energy loss through interaction with the electron gas or lattice of the detector material. These interactions are theoretically described by theory of Bethe [132], it’s modifications by Barkas [133] and Bloch [134] or theory of Firsov [135]. While parameterizations that use analytical formulas for scattering cross sections exist [136,137] and could be used to determine energy loss approximately, it is more common to employ numerical methods for this purpose.

The first mechanism is scattering on the superconducting electrons directly, or indirectly through lattice phonons. The effect is no different from the photon absorption process described in Section 2—if the scattered particle deposits enough energy to break apart a Cooper pair, the excited quasi-particles will behave the same as those excited by photons.

The second mechanism is a thermal process, first proposed for detector purposes by Scherman [138]. In this thermal process, the hotspot in the nanowire is not created by diffusion and multiplication of quasi-particles, but by direct heat exchange with the microplasma in the particle track. In these tracks, the radius of a (locally) cylindrical region at temperature at least $T_{m}$ is approximately [138,139,140]
(4)$$r_{hs}\approx \sqrt{\frac{Q}{e\pi c\rho \left(T_{m}-T_{0}\right)}},$$
where *Q* is the energy loss per unit length (assumed here to stay constant through the thin film), *c* the superconductor specific heat, ρ the density and $T_{0}$ is the temperature of the substrate. By setting the temperature $T_{m}$ equal to superconducting critical temperature $T_{c}$ and carrying out a Monte-Carlo simulation [141] for values of *Q*, we can determine that the thermal hotspot radius in NbN detectors is of the order of 50–100 nm (see Figure 4). This number is only for the thermal hotspot “core”, where superconductivity is suppressed completely. Area of the region with suppressed order parameter and increased thermal fluctuations is larger than that, as the temperature increase is $\Delta T\propto \frac{1}{r^{2}}$.

This result demonstrates that SNSPDs should work not only as high efficiency single photon detectors, but also as detectors of single charged particles. There are, however, differences: In the few eV energy ranges of photonics experiments, the size of the hotspot is a monotonously increasing function of the photon energy [36,142]. With high energy particles, this is not necessarily true and one has to expect a rolloff at high energies, because the stopping power *S* (the retarding force that slows the particles in physical media) begins decreasing [143]. Because of this, *Q* (that is effectively equal to work done by stopping power S) is mostly independent on the particle energy only for E≲ 1 keV, after which there is a non-monotonic dependence on energy, as can be seen in Figure 4. can have on the detection process of particles: In case of photons, the detection process relies on scattering of a photon into a highly energetic quasi-electron pair in the volume of the superconductor. With charged particles, there is the possibility that they penetrate through the SNSPD material with minimal energy loss and interact deeper in the substrate, causing knockoff reactions or they can emit a large amount of photons and phonons as they stop due to rapid scattering at the Bragg peak—these can be energetic enough to still trigger a detection event, but the probability also has a high-energy rolloff as the particle range increases with energy and the Bragg peak moves deeper into the substrate [144].

### 4.1. Low Energy Ion Detection

The concept of detecting low energy molecules has already found a limited use in high-performance time-of-flight mass spectrometry [145], particularly in the form of micron-sized superconducting stripline detectors [146,147,148,149], which demonstrate that parallel superconducting wires provide both, high mass resolution and limited charge state discrimination of ions [150] or lysozyme molecules—this technology is discussed in greater detail in a review by Cristiano, et al. [151]. It is also worth mentioning that it is here, where MgB_2_-based devices found applications as biomolecular detectors [152].

On the other hand, the idea of using ≲ 100 nm wide SNSPDs for light particle detection purposes is a relatively unexplored concept. Several successful attempts have already been made in detecting positive ions. The first is by Sclafani, et al. [153], where it has been demonstrated that SNSPDs are capable of detection of He^+^ ions with energies up to 1 keV with 100% detection efficiency, as shown in the device bias sweeps in Figure 5a. One of the important takeaways of their analysis, based on that of Verevkin et al. [142], is that the initial hot-spot caused by a He^+^ ion is 10–20 times bigger than that of a photon with equivalent energy, which would support the model of thermal hotspots discussed in the previous paragraphs.

Another important result is that surface adsorbates, which are usually ignored in the context of photon detection, start to play an important role in detection of low energy ions, where the detection efficiency can drop by more than three orders of magnitude as the density of surface contaminants increases, as seen in Figure 5b. This is an important consideration for application in mass spectrometry, however, it might become unimportant for detection of high energy particles in NP and HEP applications, where the penetration depth of MeV and higher energy particles is easily more than 10 μm [144].

### 4.2. Low Energy Electron Detection

Following in the vein of detection of particles in the keV-range, another important results were shown by Rosticher, et al. [154], who achieve nearly 100% detection efficiency for electrons with kinetic energies up to 20 keV. In their work, Rosticher, et al. show that a SNSPD device is capable of detecting an incoming electron even if the collision doesn’t happen in the superconducting film (Figure 6a). This supports the notion that the detection does not happen just in the superconducting nanowire, but that the substrate plays an active role in the process, which explains why the detection efficiencies in particle detection can reach 100% even though the active area covered by the superconducting meander is only ≈50% and has been confirmed by the authors through a Monte-Carlo simulation of the electron kinematics. Further, these results demonstrate the previously discussed importance of stopping power of the superconducting material and the substrate, as can be clearly seen in the loss of detection efficiency at high electron kinetic energies in Figure 6b: The energy loss of an electron in certain materials is approximately inversely proportional to the kinetic energy [155], which means that it penetrates deeper into the material before meaningful amounts of energy are transferred into the substrate. If not enough energy is lost while in proximity of the nanowire, any meaningful energy transfer to the superconducting system cannot happen and the electron will not get detected. This effects leads to degradation of detection efficiency in bare SNSPD devices, however by adding a stopping layer on top of the device, one could exploit this effect and design a detector of particles only within a certain energy range that can be tuned by varying the thickness or density of the additional material.

### 4.3. High Energy Particle Detection

In the scope of NP and HEP experiments, it is important to determine if SNSPDs are capable of detection of particles not in the few keV energy range, but particles with kinetic energies from a few MeV and up to relativistic cases, where energies will go as high as few GeV. Detection of relativistic particles with SNSPDs has not been studied in literature, but the lower end of the range was explored by Azzouz, et al. [156], who demonstrated detection of MeV α and $\beta^{-}$ particles from ^210^Po and ^42^K or ^31^Si sources, respectively. The detection efficiency was lowest for the 5 MeV α-particles, reported to be 78±18% and over the timespan of 4 days, Azzouz, et al. have not seen any degradation in noise, timing or efficiency, demonstrating good radiation hardness of NbTiN SNSPDs in this radiation environment. By using an ^241^Am source covered by thin aluminum foil, Azzouz, et al. were also able to show that their devices are insensitive to nominally 5.95 keV γ and X-ray photons (efficiency reported as 0±10%). This is an interesting result, especially in context of the previously mentioned comparison of effects of ions and photons of equivalent energies by Sclafani, et al., that deserves future study. It also has immediate positive implications on certain experiments. For example in cases of nuclear recoil and photodisintegration measurements, one uses high-energy photons [157,158] to drive this process but, for the detectors, this γ-ray beam is a considerable source of noise, especially if scattered. Having sensitivity only to lower energy photons or charged particles would greatly benefit statistics of these experiments.

### 4.4. Neutron and Dark Matter Detection

Detection of charged particles is relatively straightforward because of comparatively high scattering cross sections with the lattice and electrons of the superconductor and substrate. There might be, however, desire to use SNSPDs to detect neutral or weakly interacting particles such as neutrons or potential dark matter candidates. Working detection schemes utilizing SNSPDs are not yet developed but, for detection of neutrons, there exists a related technology that was used to demonstrate this capability: current biased kinetic inductance detectors (CBKID) [159,160,161]. The operation of CBKIDs is slightly different from SNSPDs, where one detects changes of kinetic inductance due to partial suppression of superconductivity in a large meandering superconducting stripline instead of change of DC resistance of a nanoscale meander, but the main operation principle can be transferred between the two technologies. In the works of Iizawa et al. [160] and Shishido et al. [161], a separate layer of ^10^*B*, while in work of Yoshioka, et al. [162] a more direct approach was used when detectors were fabricated out of superconducting MgB_2_. ^10^B in the devices converts neutrons into energetic ^4^He and ^7^Li ions detected using a differential readout of the CBKID delay line, which allowed for a spatially-resolved measurement of neutron flux. Because the main neutron detection mechanism is through the conversion into charged particles, one can, in principle, substitute the CBKID devices with SNSPDs and construct a high-rate and high-speed detector of neutron flux with the same characteristics as discussed in the previous sections.

The last application mentioned here is in detection of dark matter. As attempts to detect dark matter with mass above the GeV scale are so far fruitless, researchers are coming up with proposals of detection of MeV-scale dark matter in semiconductors [163,164], or even lower energies, where superconductors can be used for direct detection through dark matter-electron scattering [165,166]. As of the time of writing of this review, there is no concrete realization SNSPD dark matter search experiment, but Hochberg et al. propose use of NbN-based SNSPDs as target and sensor for ultra-light dark matter [167], citing high sensitivity and very low dark counts when compared to existing experiments.

## 5. Conclusions

Many experiments in nuclear, high energy and astrophysics have demands that can be uniquely met by the capabilities of superconducting nanowire detectors. We discussed applications of SNSPDs that go beyond just detection of low-energy photons in low-background environments and show that there exist demonstrations of the technology being robust enough for development of detectors based on SNSPDs, especially in small to medium-scale experiments. Further work needs to be done, especially on the front of systematic characterization of performance of high energy particle detection and integration into large-scale experiments, however, this effort represents opportunity for experiments that are currently near impossible with conventional technologies.

## Figures and Tables

**Figure 1 nanomaterials-10-01198-f001:**
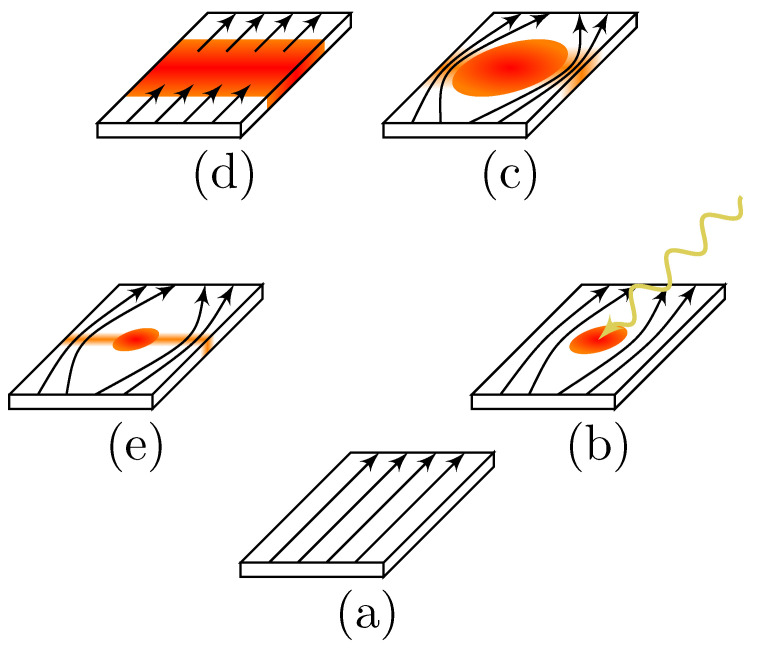
Schematic of the detection process through the hot spot formation after a photon absorption. Equilibrium superconductor is in white, orange to red depicts regions with increasingly suppressed order parameter and black lines depict the superconducting current density. Yellow arrow is a schematic depiction of an incoming single photon.

**Figure 2 nanomaterials-10-01198-f002:**
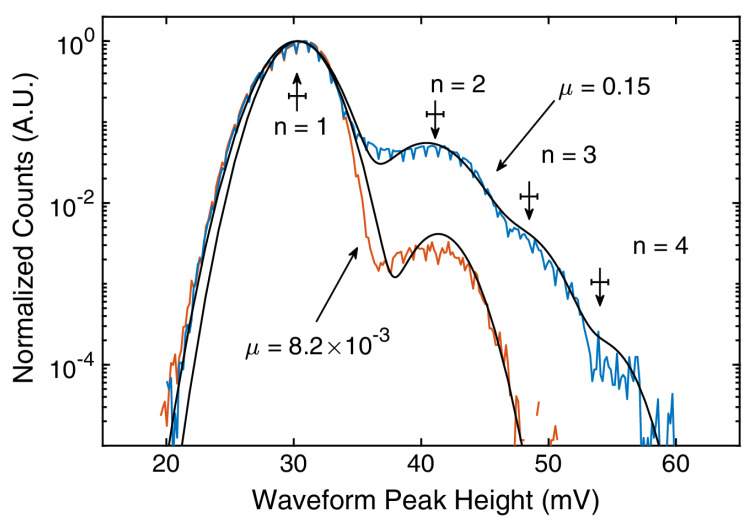
Histograms of the peak height of differentiated detection waveforms corresponding to detection of *n*-photon signal. The two curves correspond to light wave packets with different mean detected photon number μ. The arrows with corresponding error bars show predicted values of the peaks from the electro-thermal model and finite-bandwidth amplifiers. Figure reproduced from Cahal, et al., Optica **4**, 1534–1535 (2017) [108].

**Figure 3 nanomaterials-10-01198-f003:**
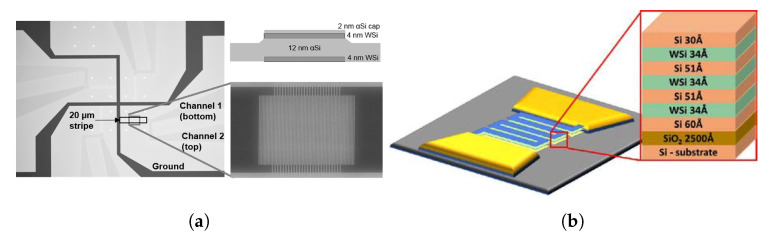
(**a**) A two SNSPD-layer device with optical image (left) of the whole structure and a schematic cross-sectional view in top-right and a SEM image of the SNSPD itself (bottom-right). Figure reproduced from Verma, et al., Appl. Phys. Lett. **108**, 131108 (2016) [114]. (**b**) Schematic of representation of a three SNSPD-layer device. Figure reproduced from Florya, et al., Low Temperature Physics **44**, 221–225 (2018) [112].

**Figure 4 nanomaterials-10-01198-f004:**
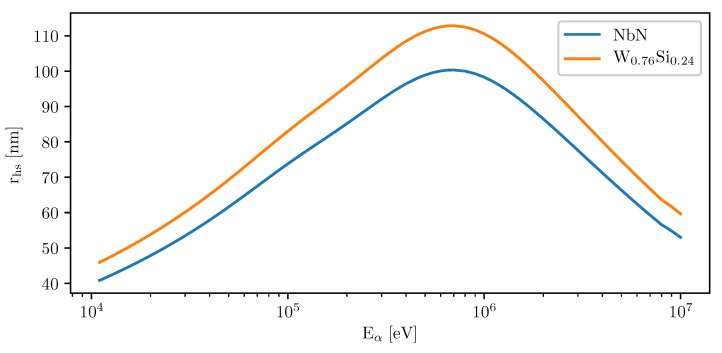
Approximate thermal hotspot radius $r_{hs}$ as a function of α-particle kinetic energy in NbN film with $T_{C}$ = 8 K and W_0.76_Si_0.24_ film with T_C_ = 3.35 K. Both films are assumed to be held at $T_{0}=\frac{T_{C}}{2}$.

**Figure 5 nanomaterials-10-01198-f005:**
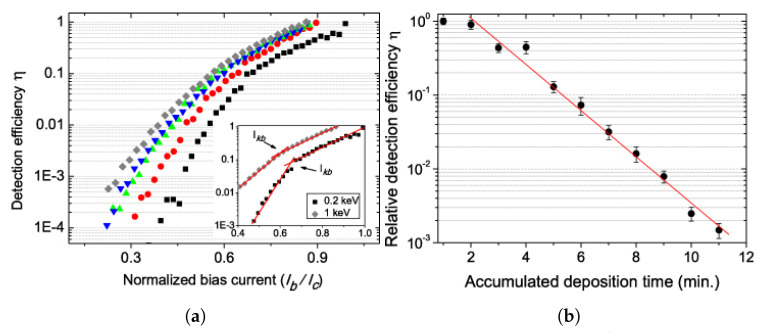
(**a**) Demonstration of capability of 100% detection efficiency of He^+^ particles with kinetic energies of 200 eV (square), 400 eV (circle), 600 eV (up-triangle), 800 eV (down-triangle) and 1000 eV (diamond). The points *I_kb_* corresponds to bias current value where the primary cause for hot-spot expansion changes from fluctuation-based into current-crowding regime. (**b**) Relative detection efficiency as a function of accumulated deposition time of neutral He atoms. Figures reproduced from Sclafani, et al., Nanotechnology **23**, 065501 (2012) [153].

**Figure 6 nanomaterials-10-01198-f006:**
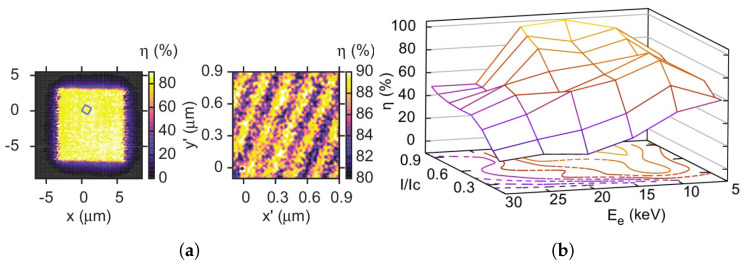
(**a**) Map of detection efficiency across the SNSPD pixel (left) under bombardment of 20 keV electrons, with a zoom in on a small part of the meander showing the non-zero detection probability outside of the superconducting meander. (**b**) A parameter sweep map that show the electron detection efficiency as a function of the reduced device bias current *I*/*I_C_* and electron kinetic energy *E_e_*. Figures reproduced from Rosticher, et al., Appl. Phys. Lett. **97**, 183106 (2010) [154].

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
