# Peer review of "Unconventional Applications of Superconducting Nanowire Single Photon Detectors"

_nanomaterials, 2020, doi:10.3390/nano10061198_

Round 1
Reviewer 1 Report
This paper gives an accurate review about SNSPDs proven applications and their future interest for high-energy physics experiments. Its extensive reference list gives the reader an up-to-date information. The paper is well structured and the prospects for high energy applications are clearly described. I think it should be published. Nevertheless there is a list of minor corrections to do.
Technical comments :
Line 19 : … can be used in.
Line 83 : …creating a normal state region the bisects …
Line 175: Similar attempts weRE made
Line 216: If needed, THESE values
Line 222: … or theory Firsov[108]. <——???
Line 222: While analytical parameterizations <—
Lines 229-230 : T0 is not defined. x does not appear in the formula.
Line 233: is OF THE order of 50–100 nm
Lines 240-241: stopping power S begins decreasing [116] and, in thin films, Q = Q(E) ∼= S(E). : it is not clear what the stopping power E is and what Q = Q(E) ∼= S(E) means.
Line 248 : BragG peak
Figure 4 caption: to be in held at
Line 251: in the form OF micron-sized
Line 253: limited charge state state discrimination
Line 275: that the detection is does not happen
Line 292: detection of not particles NOT
Lien 292 : with kinetic energies with energies
Line 320 : supercondcuting MgB2
Author Response
Dear Reviewer,
We thank you for your time reviewing our manuscript and appreciate your suggestions and valuable critics.
In the revised version we addressed all your comments and fixed all mentioned technical deficiencies of the original draft.
Our responses, together an itemized list of all changes are enclosed below.
Sincerely, V. Novosad, on behalf of all co-authors.
PS Please note, the list of changes at the very end of this message also includes the changes prompted by the second Reviewer of this work.
---
Reviewer #1:
Line 19 : … can be used in.
Line 83 : …creating a normal state region the bisects …
Line 175: Similar attempts weRE made
Line 216: If needed, THESE values
Line 222: … or theory Firsov[108]. <——???
We thank the reviewer for pointing out the spelling errors. They have been fixed in the revised manuscript.
Line 222: While analytical parameterizations <—
We were trying to convey that, even if energy loss is usually determined from numerical simulations, there exist parameterized analytical formulas for cross-sections that can be used to approximate the loss without Monte-Carlo methods. We changed the sentence to be specific about this.
Lines 229-230 : T0 is not defined. x does not appear in the formula.
The appearance of x was leftover from earlier stage of the manuscript, which used a more general form of the equation. T0 is the temperature of the heat sink, which is the substrate in this case. Mention of the variable x was removed and explanation of T0 was added.
Line 233: is OF THE order of 50–100 nm
We thank the reviewer for pointing out the text error. It was fixed in the revised version of the manuscript.
Lines 240-241: stopping power S begins decreasing [116] and, in thin films, Q = Q(E) ∼= S(E). : it is not clear what the stopping power E is and what Q = Q(E) ∼= S(E) means.
The stopping power S is the retarding force that causes particle deceleration (nominally in Newtons) and it’s the work done by this force that turns into heat Q. S (and Q) at very low energies (up to a few keV) is dependent mostly just on the mass and charge of the ion and density of the stopping material. At higher energies, this is not true and one has to account for the variation which leads to what we show in Figure 4. We changed the discussion in the text to be more clear.
Line 248 : BragG peak
Figure 4 caption: to be in held at
Line 251: in the form OF micron-sized
Line 253: limited charge state state discrimination
Line 275: that the detection is does not happen
Line 292: detection of not particles NOT
Lien 292 : with kinetic energies with energies
Line 320 : supercondcuting MgB2
Fixed. We, again, thank the reviewer for pointing out mistakes in text. They all are fixed in the revised version of the manuscript.
Itemized list of changes to the text:
- Typographical errors were removed across the text (lines 19, 83, 175, 216, 222, 233, 240, 248, 251, 253, 275, 292, 320 in the original manuscript)
- Reference to non-existent variable x in equation (4) has been removed (line 229 in the original manuscript)
- A new paragraph discussing the progress on large area detectors has been added along with appropriate references (lines 113-129 and references 56-72 in the revised manuscript)
- Added a short mention of specific use-cases for light detection in NP/HEP and reworded the opening of the section (lines 175-179 in the revised text)
- Added a paragraph on cuprates and MgB2 in SNSPDs into the materials discussion (lines 150-161 and references 89-96 in the revised manuscript)
- Reworded the explanation of energy-dependence of stopping power (270-273 in the revised manuscript)
- Added mention of MgB2 stripline low-energy ion detectors (lines 286-288 and reference 152 in the revised manuscript)
Reviewer 2 Report
Submitted review manuscript of Polokovic et al. is about the application of SNSPD for nuclear Physics (NP) and high energy physics (HEP). Since these areas are unfamiliar for the SNSPD community, the manuscript attracts the reader’s attention as the first review article on the topic. I find the article is suitable for publication in Nanomaterials, nevertheless I have several comments that should be addressed before publication:
Comments,
The NP and HEP application may require larger size SNSPD compared to standard SNSPD (10μm x 10μm). We may also use the nanowire with larger dimensions compared to standard SNSPD (width 100nm, thickness 5nm). Need discussion or comment on size and dimensions.
line 144: The authors discuss about multi-photon detection and multi-layer SNSPD. While these topics are interesting in their own right, it is not clear how these techniques will be used to NP and HEP applications. Discussion or comment required.
line 179: The authors discuss about the SNSPD in strong magnetic field up to 5T, and conclude the importance of device geometry and film deposition technology. The use of high HC2 materials such as cuprate superconductors may also be effective for the purpose. There is a need for discussion on material selection.
line 229: “where x is …”, however, there is no x in eq.(4).
line 249: The authors discuss about SNSPD for low energy molecule detection using NbN. The detection of low energy molecule has also been reported to use MgB2 ( Zen, Appl. Phys. Lett.106, 222601). I suggest to add it as a review article.
Author Response
Dear Reviewer,
We thank you for your time reviewing our manuscript and appreciate your suggestions and valuable critics.
In the revised version we addressed all comments and fixed technical deficiencies of the original draft.
Our responses, together an itemized list of all changes are enclosed below.
Sincerely, V. Novosad, on behalf of all co-authors.
PS Please note, the list of changes at the very end of this message also includes the changes prompted by the first Reviewer of this work.
-----------------------------------------
Reviewer #2:
The NP and HEP application may require larger size SNSPD compared to standard SNSPD (10μm x 10μm). We may also use the nanowire with larger dimensions compared to standard SNSPD (width 100nm, thickness 5nm). Need discussion or comment on size and dimensions.
We thank the reviewer for this important remark. The roughly 10um x 10um pixels are predominantly of that size because it is sufficient for nanophotonics experiments. But there are demonstrations of individual pixels with sizes on the order of 100um x 100um, parallel-connected multi-pixel detectors and 32x32 individual pixel arrays. Even with standard wire dimensions, this allows for detectors that can cover areas on the scale of millimeters. And, as correctly pointed out, this area can be increased even further if we move beyond photon detection and relax on the wire width and thickness.
We added discussion and an extensive list of references relevant to large-area detectors to the revised manuscript.
line 144: The authors discuss about multi-photon detection and multi-layer SNSPD. While these topics are interesting in their own right, it is not clear how these techniques will be used to NP and HEP applications. Discussion or comment required.
Our aim with this manuscript is to also reach an audience that has little experience in superconducting detectors but might benefit from using them. From an outsider perspective, it might not be readily obvious that a device called single-photon detector is also capable multi-photon detection, which could be important for applications that are outside of the expertise of the authors.
It is also related to the multi-layer SNSPD devices, which have the big benefit of being broadband detectors – this is important because most light-emitting processes interesting to NP/HEP (Cherenkov radiation, ionization, etc.) usually emit light across a relatively large part of the spectrum. Being able to collect as much signal as possible (or even better, distinguish between photons with different energies) allows for better results, be it just due to higher signal-to-noise or capability to veto false signals.
To address the Reviewer's concern , we modified the opening of section 3 and highlighted the possible uses for SNSPDs as light detectors.
line 179: The authors discuss about the SNSPD in strong magnetic field up to 5T, and conclude the importance of device geometry and film deposition technology. The use of high HC2 materials such as cuprate superconductors may also be effective for the purpose. There is a need for discussion on material selection.
It is true that if going purely by the demand of high TC or HC2, cuprates or MgB2 would be good candidates. The materials are, however, notoriously difficult to work with in thin film form and their degradation due to environmental factors and radiation is well documented in literature. Nevertheless, we do agree with the reviewer that they should be part of the discussion.
We added discussion about use of unconventional superconductors and MgB2 for SNSPDs in the text.
line 229: “where x is …”, however, there is no x in eq.(4).
We thank the reviewer for pointing this out this mistake.
The mention of x was removed from the manuscript.
line 249: The authors discuss about SNSPD for low energy molecule detection using NbN. The detection of low energy molecule has also been reported to use MgB2 ( Zen, Appl. Phys. Lett.106, 222601). I suggest to add it as a review article.
We greatly appreciate the recommendation. The reference was added to the manuscript and mentioned in the section of low-energy ion detection section.
Itemized list of all changes to the text:
- Typographical errors were removed across the text (lines 19, 83, 175, 216, 222, 233, 240, 248, 251, 253, 275, 292, 320 in the original manuscript)
- Reference to non-existent variable x in equation (4) has been removed (line 229 in the original manuscript)
- A new paragraph discussing the progress on large area detectors has been added along with appropriate references (lines 113-129 and references 56-72 in the revised manuscript)
- Added a short mention of specific use-cases for light detection in NP/HEP and reworded the opening of the section (lines 175-179 in the revised text)
- Added a paragraph on cuprates and MgB2 in SNSPDs into the materials discussion (lines 150-161 and references 89-96 in the revised manuscript)
- Reworded the explanation of energy-dependence of stopping power (270-273 in the revised manuscript)
- Added mention of MgB2 stripline low-energy ion detectors (lines 286-288 and reference 152 in the revised manuscript)